# Healthcare waste generation and quantification in public health centres in Addis Ababa, Ethiopia

**Menelik Legesse Tadesse** *

Menelik II Medical and Health Science College, Addis Ababa, Ethiopia

* menelik3rd@yahoo.com

## Abstract

### Background

Healthcare waste produced in healthcare activities entails higher risk of infection and injuries than municipal waste. In developing countries healthcare waste has not received much attention and has been disposed of together with municipal wastes. Modern method of disposal of healthcare waste have been introduced to most healthcare institutions mismanagement and increased in production in public health centres in Ethiopia is important issues. The aim of the study was to assess the type of healthcare waste generation and quantification in selected public health centres in Addis Ababa, Ethiopia.

### Methods

An institution based cross-sectional study were conducted from January to February 2018. Fifteen health centres in Addis Ababa City Administration were selected for this study. Data were collected by using by different color plastic bags (Black plastic bags for non-hazardous wastes, Yellow plastic bags for hazardous wastes and Yellow safety box for needles and Red bags for pharmaceutical wastes and toxic wastes). The collected wastes were measured by weighing scale and were written to data entry sheet. To assure the data quality calibration of weighing scale was made by the standard weight every morning. EPI INFO TM7 and IBM SPSS were used for data entry, cleaning and analysis.

### Results

The mean healthcare waste generation was 10.64+5.79Kg/day of which 37.26% (3.96 +2.20Kg/day) was general waste and 62.74% (6.68+4.29) was hazardous waste from the studies health centres. Total hazardous waste; sharps, infectious, pathological and pharmaceutical wastes constitutes mean (±SD) 0.97 ±1.03, 3.23 ± 2.60, 2.17±1.92 and 0.25 ±0.34 kg/day respectively. Healthcare waste 29.93% and 0.32% were generated from delivery and post-natal case team and nutrition and growth monitoring case team respectively. The annual mean+ SD of healthcare waste generation rate per health centres were 3807.53+ 2109.84 Kg/year.

**Data Availability Statement:** UNISA INSTITUTIONAL REPOSITORY https://uir.unisa.ac.za/handle/10500/26614 URL for abstract version

https://uir.unisa.ac.za/bitstream/handle/10500/26614/thesis_menelik_legesse_tadesse.pdf?sequence=1&isAllowed=y URL for full thesis version

**Funding:** The authors received no specific funding for this work.

**Competing interests:** The authors have declared that no competing interests exist.

## Conclusion

The finding in this study showed there was an increased in hazardous healthcare waste in amount as compared to the WHO standard 85% non-hazardous waste and 10% hazardous waste and 5% toxic wastes. The healthcare waste management practices about segregation, collection, transportation and disposal at the source is crucial to decrease in quantity. Generally unselective handling and disposal of healthcare wastes is a concern.

## Background

Healthcare waste (HCW) produced in the course of health care activities entails a higher risk of infection and injuries than municipal waste. Improper treatment of this waste poses serious risks of disease transmission to waste handlers, health workers, patients, community in general and to environment [1]. Different kinds of therapeutic procedures (surgery, delivery, resection of gangrenous organs, autopsy, biopsy, para-clinical tests, and injections) are carried out in healthcare facilities and result in the production of hazardous substances, including pathological and infectious wastes, sharp objects and chemical materials these healthcare wastes may carry germs of disease such as hepatitis B and AIDS [2]. Approximately 80% of the general waste is mixed with a hazardous component [3]. It amplifies the rate of waste generation and increases threats to the safety of health workers and patients [4].

In developing countries, healthcare waste has not received much attention and has been disposed of together with municipal waste [5]. In Ethiopia, improper healthcare waste management is alarming and poses a serious threat to public health [6]. The risk of healthcare waste and its management has become a global cause of concern. Healthcare waste production at hospitals and its management are important issues worldwide [7]. Since the mid-1990s the world has experienced a dramatic increase in the amount of hazardous waste generated. The quantity of HCWs has increased in many nations due to the spread of the coronavirus disease (COVID-19), waste challenge and increasing urgency to address environmental sustainability offer an opportunity to strengthen systems to safely and sustainably reduce and manage health care waste [8]. Healthcare waste has done much damage to the environment and public health and has been the cause of a high death toll from waste-related diseases [9]. A comparison of waste classification between China and the European Union, Japan, and the United States of America found that incinerator workers and people living near incinerators had significantly higher levels of dioxins, furans and hydrocarbon compound in their blood and urine [10].

Health problems and removing potential risks to human health, health services inevitably create waste that can pose a health hazard in itself [11]. Studies have been conducted on the generation rate and composition of healthcare waste in Africa examined the healthcare waste status in selected healthcare facilities in Lagos State, Nigeria, West Gojam, Ethiopia and Addis Ababa, Ethiopia [6, 12, 13]. For example, in Mauritius, the percentage of healthcare waste has increased significantly since the 1990s due to population growth, increased number and size of healthcare facilities and the use of disposable medical products [14].

The majority of the problems are associated with an exponential growth in the healthcare sector together with low or non- compliance with guidelines and recommendations [11]. A cross-sectional survey conducted in Ethiopia Oromia region Jimma Zone, showed that out of 174 health workers exposed to human immune deficiency virus risk, 105(60.3%) sustained needle prick/cut by sharps, 77(44.3%) were exposed to blood and 68(39.1%) to patients' body fluid [15].

There is growing public concern about HCW in Ethiopia, particularly in Addis Ababa [16]. The concern is about the lack of appropriate HCW segregation, selection, handling, storage, transport, treatment and final disposal. A large proportion of HCW consists of solid waste, which cause health hazards and environmental hazard. Between 2011 and 2022, the Addis Ababa City Administration Health Bureau built more than 75 health centres (HC) and two (2) referral hospital. In addition, the expansion of infrastructure and services to hospitals and health centres is also critical for the production of healthcare waste. This motivated the researcher to conduct this study to assess the generation and quantification of healthcare waste (HCW) in health facilities in Addis Ababa Ethiopia.

## Materials and methods

### Study setting, design and population

The study setting was Addis Ababa, the capital city of Ethiopia, with the estimated population of 3,945,000 with a projected annual growth rate of 2.5% [17]. Of the population, 1,861,000 (47.2%) were male and 2,084,000 (52.8%) were female. It is the largest and most populous city in the country. The city has three layers of administration, the city administration at the top, 11 sub-cities administration in the middle and 120 Weredas (Disrticts) at the bottom [18]. An institutional based cross- sectional study was conducted. There were 101 functional public health centres found in Addis Ababa City Administration Health Bureau. Fifteen health centres were selected for this study and all case teams/departments were the study unit for this study.

### Inclusion and exclusion criteria

In this study solid healthcare waste; Non-hazardous waste (general waste) and hazardous wastes (sharps, infectious, pathological and pharmaceutical) were included. Liquid wastes were excluded from this study.

### Sample size and sampling techniques

A mean sample size estimating for a very large population formula was used to calculate the sample size [19]; considering an assumption of proportion 95% level of confidence and 6.37% marginal error. This was done by referring from previous studies mean generation: 9.61 kg per day; SD 3.28 kg per day over 100 measurements in 10 health centres [6]. Seven consecutive days sample measurement was done in fifteen health centers (Fig 1). Pre-prepared data entry sheet for the daily measurement of the amount of healthcare waste rate were used.

### Data collection tool and procedure

The researcher visited selected health centres to measure the exact amount of healthcare waste types and generation per day. All case teams/departments in the health centres were examined for the generation and quantification of healthcare waste. Data was collected by means of weighing scale measurement. Data entry sheet was prepared for the daily measurement of the amount of healthcare waste rate and daily patient visit at each health centers for seven (7) consecutive days by trained data collectors. Data was collected for every 24 hours from each case teams/departments in different color-coded polyethylene bags. The bags color was yellow, red and black. Yellow bags contained hazardous healthcare waste including yellow safety boxes for sharp waste. Black bags contained for non-hazardous (non- risk) waste. Red bags contained for pharmaceutical wastes and radioactive hazardous waste. Each plastic bag was labeled by BANNER brand sticker showed the date and time started, category of waste and room where

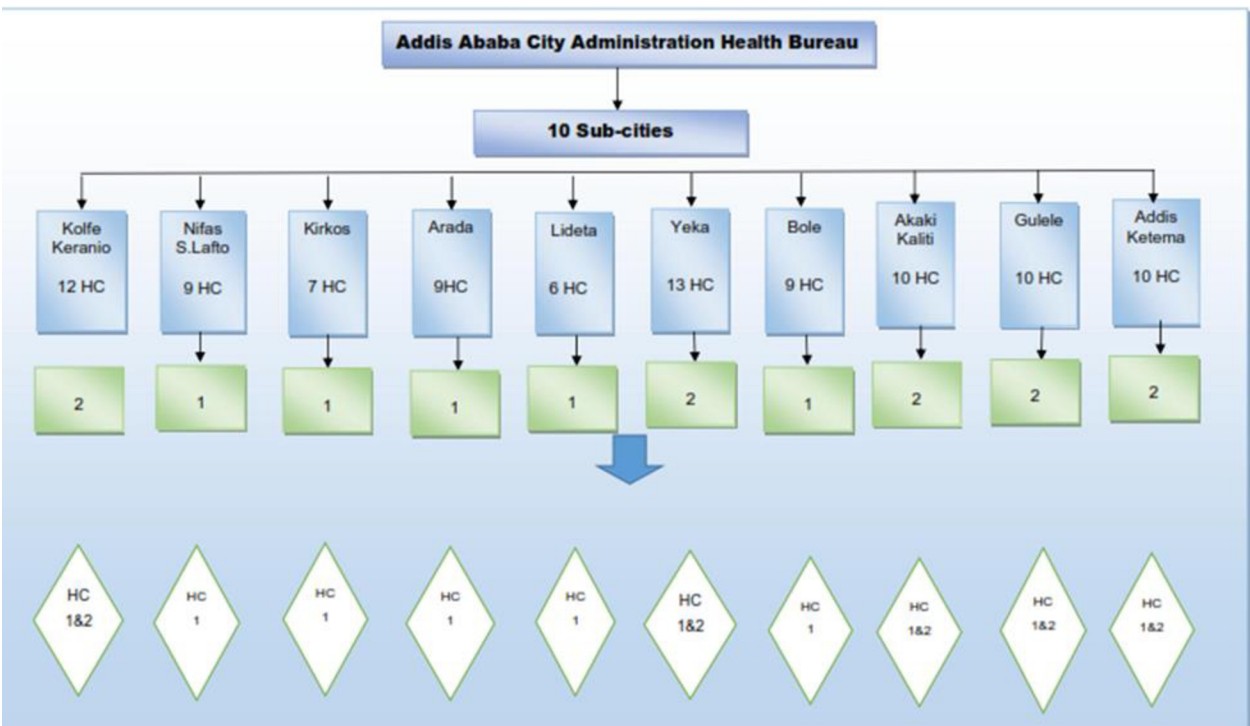

**Fig 1. Number of health centres selected from ten sub cities of Addis Ababa City Administration, February 2018.**

waste was collected. Total waste per day was measured at each study unit by removing the plastic bags every morning at 7:30 am in the morning and its weight was measured every day at 09:00 A.M using weighing scale. At least nineteen (19) case teams were chosen; Pharmacy, Laboratory, Focus Antenatal Care (FNAC), Delivery and post-natal, Tuberculosis (TB) and Leprosy, Expanded program for Immunization (EPI), Family planning, HIV/AIDS Counselling and Testing (HTC), Antiretroviral Treatment (ART), Medical recording, Nutrition and growth monitoring (NGM), Abortion procedures, Health Management Information System (HMIS), In-patient, Laundry and Adolescence and youth, Emergency Injection and Dressing room and Out Patient Department (OPD).

## Data quality control

Fifteen data collectors who graduated from a secondary school with Grade 10 certificates were used for measurement of healthcare waste. Eight supervisors who were BSc graduates in Environmental Health or related fields assisted the principal investigator with the data collection. The researcher prepared training manual on how to address the data collection check list and how to weigh the healthcare waste. The researcher gave the data collectors and supervisors three days of training in data collection and the data-collection tools as well as the relevant protocols and precautionary measures not to contaminate themselves and others and field visit. The researcher verified data and controlled the study. The supervisors calibrated the measuring balances and facilitated the collection. Pre-test was done at two health centers of the same structure other than the sampled areas.

A digital weighing scale model Sartorious Basic Type BA 6100 and electronic compact Balance Model EPB-10001 L digital were used to measure healthcare waste at all the selected healthcare facilities. Calibration was done by using a standard of 20g, 100g, 500g and 1000g

weighting objects every morning before the actual measurement started. The standard value was recorded for comparison to the daily activities. Daily onsite supervision was made by the supervisors and principal investigator during the actual measurements and tools assuring ethical issue and respondents assuring an animosity.

## Data processing and analysis

To determine healthcare waste generation rate and quantification data from data entry sheet entered to Microsoft Excel 2016 in daily basis. EPI- INFO TM 7 and IBM SPSS 20 were used for data entry, cleaning and analysis. Data analysis was performed separately for each of the health centres which were grouped by category of ownership. Spearman's correlation; means; Standard Deviation (SD); frequencies; percentages, and graphs were used.

## Ethical consideration

Ethical approval and clearance were obtained from the Higher Degrees Committee, Department of Health Studies, University of South Africa and Addis Ababa City Administration Health Bureau to conduct the study. The researcher has gotten written and verbal consent to begin the study in the respected health centres directors and case teams. Data collectors were provided data collection manual and trained to use Personal Protective Equipment while handling healthcare wastes. The researcher and supervisors were made alert about the provision of medical assistance for sharps and needle prick injuries.

## Results

### Patient flow and daily HCW generation in the study health centres

A total of 13,897 patients visited the selected health centres on the seven (7) consecutive days of data collection. Of these, 1,765 (12.7%) and 1,527 (10.99%) visited Meshoalekia and Filipos health centres, and 474 (3.41%) and 466(3.35%) visited Woreda 9 and Korea Zemachoch health centres, respectively. The mean (±SD) patients per day in all the selected health centres was 132.35±60.621 (Table 1).

The mean (±SD) HCW generation rate was 10.64 ± 5.790 kg/day, of which 3.96 ± 2.017 kg/day (37.26%) was general waste and 6.68 ± 4.293 kg/day (62.74%) was hazardous waste. A high amount of HCW per day was generated at Filipos and Yeka health centres 26.90 kg/day and 16.96 kg/day, respectively. A small amount of HCW generation was recorded at Woreda 9 and Korea Zemachoch health centres 4.71 kg/day and 5.25 kg/day, respectively (Table 3). All the health centres operated daily and were open for 24 hours and offered services, but Filipos HC had more patients than the others. In this study revealed that Filipos health centre had the highest HCW generation rate with an average of 0.123 kg/patient/day (Table 1).

### Types and amount of HCW generation

The results for HCW collected weekly from the study health centres varied in amount of HCW generation. The average value for HCW in each health centre and standard deviation as error bar. The findings indicate significant variations in the HCW generation rates (Fig 2).

The types of hazardous waste generated by the study health centres were sharps, infectious, pathological (placenta and blood) and pharmaceutical. The mean (±SD) generation rate of sharps, infectious, pathological and pharmaceutical waste in each health centre was 0.97± 1.031 (14.63%), 3.23± 2.603 (48.72%), 2.17± 1.917 (32.73%) and 0.26± 0.342 (3.58%) kg/day, respectively (Table 2).

**Table 1. Number of patients visited and daily HCW generation rate in the study public health centres, Addis Ababa City Administration, February 2018.**

| Name of health centre | Total patients visited health centre on 7 days (Observed) | Mean patients visited health centre | Healthcare waste (HCW), Kg/day | | | |
|---|---|---|---|---|---|---|
| | | | Total HCW in 7 days (observed) | Mean of HCW Mean ($\pm$ SD) | Mean of general waste (%) | Mean of hazardous waste (%) |
| Kolfe | 1503 | 214.71 | 67.44 | 9.63$\pm$ 15.328 | 1.45 (15.31%) | 8.19 (84.69%) |
| Filipos | 1527 | 218.14 | 188.30 | 26.90$\pm$31.341 | 8.00 (29.75%) | 18.90 (70.25%) |
| Meshoalekia | 1765 | 252.14 | 53.88 | 7.69$\pm$11.041 | 3.79 (49.28%) | 3.90 (50.72%) |
| Teklehaymanot | 694 | 99.14 | 39.48 | 5.63$\pm$6.077 | 2.38 (42.14%) | 3.26 (57.86%) |
| Woreda 1 | 685 | 97.86 | 110.11 | 15.70$\pm$19.070 | 6.60 (41.97%) | 9.10 (58.03%) |
| Kella | 620 | 88.57 | 66.30 | 9.47$\pm$11.382 | 4.098 (43.27%) | 5.37 (56.73%) |
| Saris | 784 | 112.00 | 94.41 | 13.49$\pm$10.446 | 2.92 (21.66%) | 10.57 (78.34%) |
| Korea Zemachoch | 466 | 66.57 | 36.57 | 5.25$\pm$7.489 | 2.44 (46.77%) | 2.80 (53.23%) |
| Yeka | 872 | 124.57 | 118.71 | 16.96$\pm$17.963 | 6.86 (40.42%) | 10.10 (59.58%) |
| Goro | 1498 | 214.00 | 68.75 | 9.82$\pm$13.602 | 2.98 (30.35%) | 6.84 (69.65%) |
| Millennium | 689 | 98.43 | 40.75 | 5.82$\pm$8.419 | 2.85 (49.0%) | 2.97 (50.99%) |
| Woreda 9 | 474 | 67.71 | 32.96 | 4.71$\pm$6.303 | 1.87 (39.81%) | 2.83 (60.19%) |
| Michew | 690 | 98.57 | 58.35 | 8.33$\pm$7.752 | 2.41 (28.97%) | 5.92 (71.03%) |
| Sheromeda | 901 | 128.71 | 65.27258 | 9.32$\pm$15.958 | 5.29 (56.77%) | 4.03 (43.23%) |
| Arada | 729 | 104.14 | 76.478 | 10.93$\pm$15.615 | 5.47 (50.07%) | 5.45 (49.93%) |
| Overall mean | 926.47 | 132.35 | 74.51 | 10.64 | 3.96(37.26%) | 6.68 (62.74%) |
| SD | 424.35 | 60.621 | 40.532 | 5.790 | 2.017 | 4.293 |

In most of the health centres, the generation of hazardous HCW was high. For example, Filipos HC generated 18.89 kg/day; Saris HC generated 10.57 kg/day, and Yeka HC generated 10.11 kg/day. The lowest amounts of hazardous HCW generated were from Korea Zemachoch HC (2.81 kg/day), Woreda 9 HC (2.83 kg/day) and Millennium HC (2.97 kg/day) (Table 2).

The composition and generation of hazardous HCW in the study health centres were infectious and pathological waste comprised 82% of the hazardous waste; Sharps, Infectious, Pathological and pharmaceutical wastes were 15%, 49%, 33% and 4% respectively (Fig 3).

### Daily HCW generation rate in different case teams

In different case teams, the HCW generation rate varied. The mean ($\pm$SD) HCW generation rate in each section was 10.63$\pm$5.795 kg/day. Increased amounts of HCW (29.93%) were generated from delivery and postnatal case teams while less (0.32%) HCW was generated from NGM case teams (Table 3).

### Estimated annual HCW generation rate

The estimation of HCW generation rate per year can be calculated in two ways. Firstly, by the annual patient flow and mean HCW generation rate per patient per day (the assumption was each patient who visited the health centre might generate the same amount of HCW throughout the year).

**Total HCW Generation per year**
= Mean HCW generation in kg per patient per day*Number of patients flow in a year

Secondly, by using the HCW generation rate per day (kg/day) and number of days in the

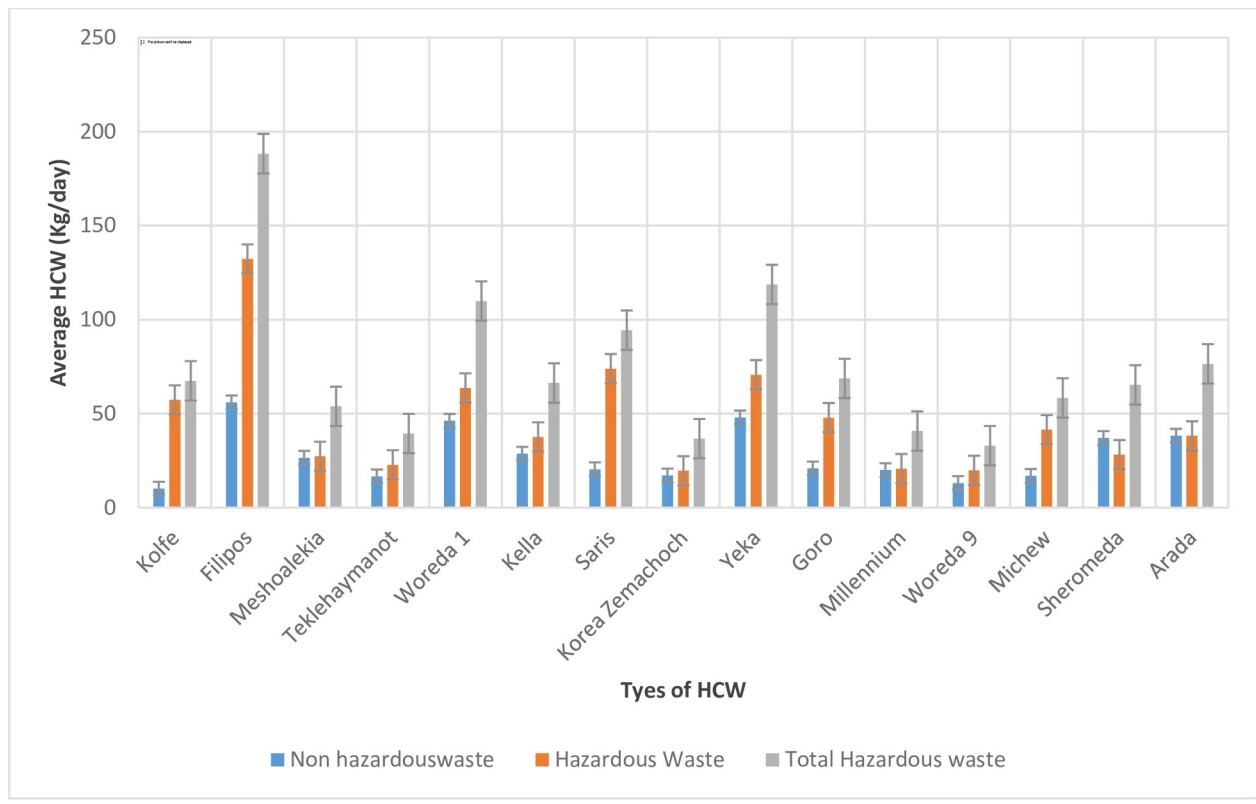

**Fig 2. Error bar of average HCW generation rates in the study public health centres, Addis Ababa City Administration, February 2018.**

**Table 2. Distribution of type and amount of daily hazardous HCW generation rate in the study public health centres, Addis Ababa City Government, February 2018.**

| Name of health centre | Sharps Kg/day | Infectious Kg/day | Pathological Kg/day | Pharmaceutical Kg/day | Total hazardous waste Kg/day |
|---|---|---|---|---|---|
| Kolfe | 0.73 | 0.19 | 5.77 | 0.37 | 7.06 |
| Filipos | 4.03 | 11.59 | 3.28 | 0 | 18.89 |
| Meshoalekia | 0.71 | 2.55 | 0.65 | 0 | 3.90 |
| Teklehaymanot | 0.78 | 1.11 | 1.37 | 0 | 3.26 |
| Woreda 1 | 0.51 | 4.07 | 4.30 | 0.21 | 9.13 |
| Kella | 0.399 | 2.67 | 2.09 | 0.21 | 5.37 |
| Saris | 2.51 | 4.71 | 2.83 | 0.51 | 10.57 |
| Korea Zemachoch | 0.64 | 1.91 | 0.26 | 0 | 2.81 |
| Yeka | 1.06 | 4.03 | 4.37 | 0.64 | 10.11 |
| Goro | 0.28 | 1.88 | 4.68 | 0 | 6.84 |
| Millennium | 0.68 | 1.96 | 0.33 | 0 | 2.97 |
| Woreda 9 | 0.36 | 1.94 | 0.54 | 0 | 2.83 |
| Michew | 0.69 | 3.24 | 1.78 | 0.43 | 6.14 |
| Sheromeda | 0.396 | 3.14 | 0.26 | 0.23 | 4.03 |
| Arada | 0.82 | 3.42 | 0 | 1.21 | 5.46 |
| Average | 0.97 | 3.23 | 2.17 | 0.25 | 6.63 |
| SD | 1.031 | 2.603 | 1.917 | 0.34 | 4.274 |

**Table 3. Distribution and daily HCW generation rates by point of source in the study public health centres, Addis Ababa City Government, February 2018.**

| Case teams | Healthcare waste (Kg/day) Mean+ (SD) | Percent | Mean rank* |
|---|---|---|---|
| OPD (Outpatient department) | 0.59+0.390 | 5.63 | 194.8 |
| Pharmacy | 0.99+0.636 | 9.28 | 225.43 |
| Laboratory | 1.76+1.094 | 16.59 | 255.53 |
| Emergency and triage | 1.085+0.893 | 10.21 | 222.60 |
| Injection and dressing | 0.64+0.537 | 5.99 | 181.1 |
| FNAC (Focus antenatal care) | 0.25+0.227 | 2.39 | 141.7 |
| Delivery and post-natal | 3.18+2.557 | 29.93 | 247.8 |
| TB and Leprosy | 0.18+0.173 | 1.66 | 119.73 |
| EPI (Expanded program for Immunization) | 0.54+0.607 | 5.07 | 180.23 |
| Family planning | 0.32+0.300 | 3.05 | 156.07 |
| HTC (HIV Testing and counselling) | 0.18+0.125 | 1.70 | 123.87 |
| ART (Anti-Retroviral Treatment) | 0.34+0.334 | 3.22 | 142.63 |
| Medical recording | 0.21+0.145 | 1.97 | 134.33 |
| NGM (Nutrition and growth monitoring) | 0.03+0.068 | 0.32 | 54.03 |
| Abortion procedures | 0.038+0.086 | 0.36 | 53.17 |
| HMIS (Health Management and Information System) | 0.041+0.081 | 0.39 | 56.50 |
| In-patient | 0.077+0.119 | 0.72 | 73.73 |
| Laundry | 0.14+0.111 | 1.34 | 109.43 |
| Adolescence and youth | 0.019+0.057 | 0.18 | 44.6 |
| Mean | 10.63 | | |
| SD | 5.795 | | |

year (the assumption was the mean of HCW per day might represent throughout 365 days).

$$\text{Total HCW generation per year} = \text{Mean HCW generation in Kg/day} * 365$$

The mean (±SD) patient flow per day per health centre was 132.35±60.621. The mean (±SD) HCW generation rate per health centre was per day or 10.63 ± 5.796 kg/day (Table 4).

The annual mean (±SD) of HCW generation rate per health centre was 3870.53 ±2109.84kg/year using the first method and 3881.14±2195.01 kg/year using the second assumption. There was a slight variation of the annual HCW generation rate in both assumptions (Table 4).

## Patient flow and HCW generation comparison

The patient flow and HCW generation rate and types, such as general and hazardous waste (sharps, infectious, pathological and pharmaceutical waste), among the study health centres were compared using the Kruskal-Wallis test to check for the presence of significant differences among their values. In this study there was no statistically significant difference for the mean patient flow ($x2 = 14.00$, p-value = 0.450), the mean general waste ($x2 = 22.631$, p-value = 0.067), and hazardous healthcare waste ($x2 = 9.421$, p-value = 0.803) (Table 5).

However, there was statistically a significant difference for mean of non- hazardous healthcare waste ($x2 = 35.819$, p-value = 0.001) among the study health centres (Table 5).

## Patient and HCW generation comparison

The extent or strength of linear relationships between the number of patients and amount of HCW generation rate was checked using the Spearman's rank correlation coefficient ($r_s$) in all

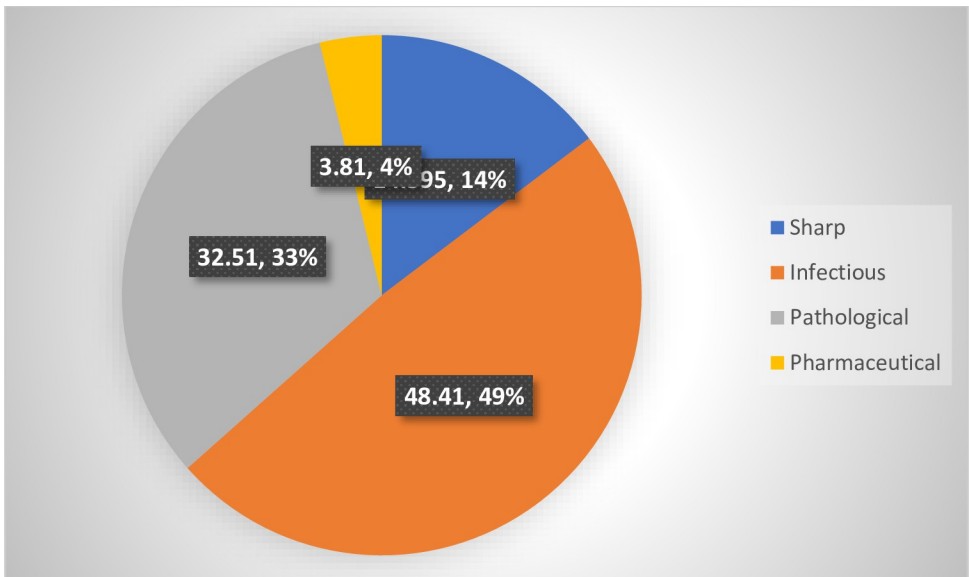

**Fig 3. Composition, contribution and generation of hazardous HCW in the study public health centres, Addis Ababa City Administration, February 2018.**

the health centres. The Spearman's rank correlation coefficient showed a positive linear relationship: as the number of patients increased, HCW also increased in all the study health centres. A strong linear relationship was observed at Filipos and Meshoalekia health centres: the Spearman's correlation coefficient was 0.964 and 0.964, respectively which is far from a perfect

**Table 4. Estimated total HCW generation rate per year in the study public health centres, Addis Ababa City Government, February 2018.**

| Name of health centre | Patient flow in 2018 | Patient flow per day 2018 | Mean HCW kg/day | Mean HCW g/pat/day | *Total HCW kg/year | **Total HCW kg/ year |
|---|---|---|---|---|---|---|
| Kolfe | 78371 | 214.71 | 9.63 | 44.85 | 3514.95 | 3514.94 |
| Filipos | 79622 | 218.14 | 26.90 | 123.31 | 9818.5 | 9818.19 |
| Meshoalekia | 92032 | 252.14 | 7.7 | 30.53 | 2810.5 | 2809.74 |
| Teklehaymanot | 36187 | 99.14 | 5.64 | 56.89 | 2058.6 | 2058.68 |
| Woreda 1 | 35718 | 97.86 | 15.73 | 160.74 | 5741.45 | 5741.31 |
| Kella | 32329 | 88.57 | 9.47 | 106.93 | 3456.55 | 3456.94 |
| Saris | 40880 | 112 | 13.49 | 120.42 | 4923.85 | 4922.77 |
| Korea Zemachoch | 24299 | 66.57 | 5.22 | 78.48 | 1905.3 | 1906.99 |
| Yeka | 45469 | 124.57 | 16.96 | 136.13 | 6190.4 | 6189.70 |
| Goro | 78110 | 214 | 9.82 | 45.90 | 3584.3 | 3585.25 |
| Millennium | 35926 | 98.43 | 5.82 | 59.14 | 2124.3 | 2124.66 |
| Woreda 9 | 24716 | 67.71 | 4.71 | 69.54 | 1719.15 | 1718.75 |
| Michew | 35979 | 98.57 | 8.34 | 84.56 | 3044.1 | 3042.38 |
| Sheromeda | 46981 | 128.71 | 9.32 | 72.44 | 3401.8 | 3403.30 |
| Arada | 38012 | 104.14 | 10.93 | 104.91 | 3989.45 | 3987.84 |
| Average | 48308.73 | 132.35 | 10.65 | 86.32 | 3885.55 | 3885.43 |
| SD | 22126.89 | 60.621 | 5.794 | 37.731 | 2114.74 | 2114.50 |

*Total HCW generation rate in kg per year = HCW generation rate in kg per day * 365

**Total HCW generation rate in kg per year = (HCW generation rate in grams per patient per day * Number of annual patients flow)/1000

**Table 5. Comparison of patient flow and HCW generation rate and types among the study public health centres, Addis Ababa City Government, February 2018.**

| Name of health centre | Mean rank | | | |
|---|---|---|---|---|
| | Patient flow | Total HCW | Non-hazardous HCW | Hazardous HCW |
| Kolfe | 214.7 | 9.63 | 1.45 | 8.19 |
| Filipos | 218.1 | 26.90 | 8.00 | 18.90 |
| Meshoalekia | 252.1 | 7.70 | 3.79 | 3.90 |
| Teklehaymanot | 99.1 | 5.63 | 2.38 | 3.26 |
| Woreda 1 | 97.9 | 15.70 | 6.60 | 9.10 |
| Kella | 88.6 | 9.47 | 4.10 | 5.37 |
| Saris | 112 | 13.49 | 2.92 | 10.57 |
| Korea Zemachoch | 66.6 | 5.25 | 2.44 | 2.80 |
| Yeka | 124.6 | 16.96 | 6.86 | 10.10 |
| Goro | 214 | 9.82 | 2.98 | 6.84 |
| Millennium | 98.4 | 5.82 | 2.85 | 2.97 |
| Woreda 9 | 67.7 | 4.71 | 1.87 | 2.83 |
| Michew | 98.6 | 8.34 | 2.41 | 5.92 |
| Sheromeda | 128.7 | 9.32 | 5.29 | 4.03 |
| Arada | 104.1 | 10.93 | 5.47 | 5.45 |
| Chi-square | 14 | 22.631 | 35.819 | 9.421 |
| Asymp. Sig. | 0.450 | 0.067 | 0.001 | 0.803 |

linear relationship at Spearman's correlation coefficient value ($r_s = 1$). A strong linear relationship was not observed at Saris, Kolfe, Teklehaymanot and Korea Zemachoch health centres: 0.126, 0.321, 0.342, and 0.342, respectively, which is far from a perfect linear relationship at Spearman's correlation coefficient value ($r_s = 1$) (Table 6).

## Score of hazardous and non-hazardous waste HCW

Hazardous HCW and non-hazardous HCW had different lower scores and hazardous HCW was higher. The median for hazardous HCW was higher than for non-hazardous waste. The

**Table 6. Relation of patient and healthcare waste generated in the study health centres, Addis Ababa City Administration, February 2018.**

| Name of health centre | Spearman's rank correlation coefficient ($r_s$) |
|---|---|
| Kolfe | 0.321 |
| Filipos | 0.964 |
| Meshoalekia | 0.964 |
| Teklehaymanot | 0.342 |
| Woreda 1 | 0.559 |
| Kella | 0.631 |
| Saris | 0.126 |
| Korea Zemachoch | 0.342 |
| Yeka | 0.643 |
| Goro | 0.607 |
| Millennium | 0.893 |
| Woreda 9 | 0.357 |
| Michew | 0.571 |
| Sheromeda | 0.607 |
| Arada | 0.429 |

first quartile (Q1) was equal to 47.32 kg/7days to the total HCW, about 25% of the total HCW was lower than 47.32 kg/7 days and about 75% was above 47.32 kg/7 days. Regarding nonhazardous HCW, the first quartile (Q1) was equal to 17.0 kg/7 days about 25% of nonhazardous HCW was lower than 17.0 kg/7 days. The total HCW showed a lower cut-off -9.87 and an upper cut-off 75.57 kg/7 days. The hazardous HCW also had a lower cut-off -28.1 kg/7 days and an upper cut-off 32.41 kg/7 days (Fig 4).

## Relationship between HCW generated in kg/day and the daily number of patients

Relationship between HCW daily amounts generated in kg/day and the daily number of patients visited was compared by a scatter plot in study health centres. A linear trend was evident between the amount of total HCW generation and total number of patients (statistically significant, $P<0.067$; $R2 = 0.135$). Therefore, the number of patients that visited the health centres daily could be used as a predictor of HCW generation rates in the health centres. This $R2$ also showed a moderate linear relationship between the number of patients that visited the health centres and the amount of HCW generated. In particular, 13.5% of the variability among the observed values of HCW generation in the 7 days of HCW measurement was explained by the linear relationship between the total number of patients that visited the health centres and generation of HCW (Fig 5).

## Discussion

In this study the mean HCW generation rate was $10.64 \pm 5.79$ kg/day, of which 37.26% ($3.96 \pm 2.017$kg/day) was general waste and 62.74% ($6.68 \pm 4.293$ kg/day) was hazardous waste other similar study done in Bench Maji Zone Ethiopia, health centres showed out of the total HCW generated, more than half (57.9%) was general or non-risk HCW, and the remaining 42.1% was hazardous healthcare waste [15]. In both studies the amount of HCW generated in the studied health centers was different from WHO report in health facility setting that general waste (non-hazardous waste) was 85% while hazardous was 15% [20]. The reason for producing a large amount of hazardous HCW might be where there is no or ignoring the strategies in the facility level to facilitate proper segregation of waste at source. The study done in Addis Ababa health centres confirmed the segregation at source were weak 52.86%, 28.57% and 10% rated good, very good and excellent respectively [21].

   In this study of the total hazardous waste; sharps, infectious, pathological and pharmaceutical wastes constitutes with mean (±SD) $0.97 \pm 1.03$, $3.23 \pm 2.60$, $2.17 \pm 1.92$ and $0.25 \pm 0.34$ kg/day respectively in all health centers. The study done in Uganda, East Africa hazardous healthcare waste produced was 20.78Kg/day, 3.13Kg/day and 3.24Kg/day for sharp, anatomical waste and pharmaceutical waste respectively [22]. In this regard the health facilities need to prepare and distribute healthcare waste containers having the capacity of 0.024 m$^3$ all case team in the health centres.

   The mean daily generation of HCW from the studied health centres were 0.08Kg/Patient /day and the amount of healthcare waste generation rate in different case teams in this study varied high quantity 29.93% of HCW was found in delivery and postnatal case team whereas fewer amounts 0.18% of healthcare waste was generated at adolescence and youth case team. Similar study done in Adama, Ethiopia showed the average daily generation of waste from health centers were ranged from 0.02 to 0.03 Kg/Patient /day and the highest quantity of HCW in all studied health centers was attributed to delivery rooms, which accounted for 39.8% of HCW, and the least amount was recorded at the voluntary counseling and testing department,

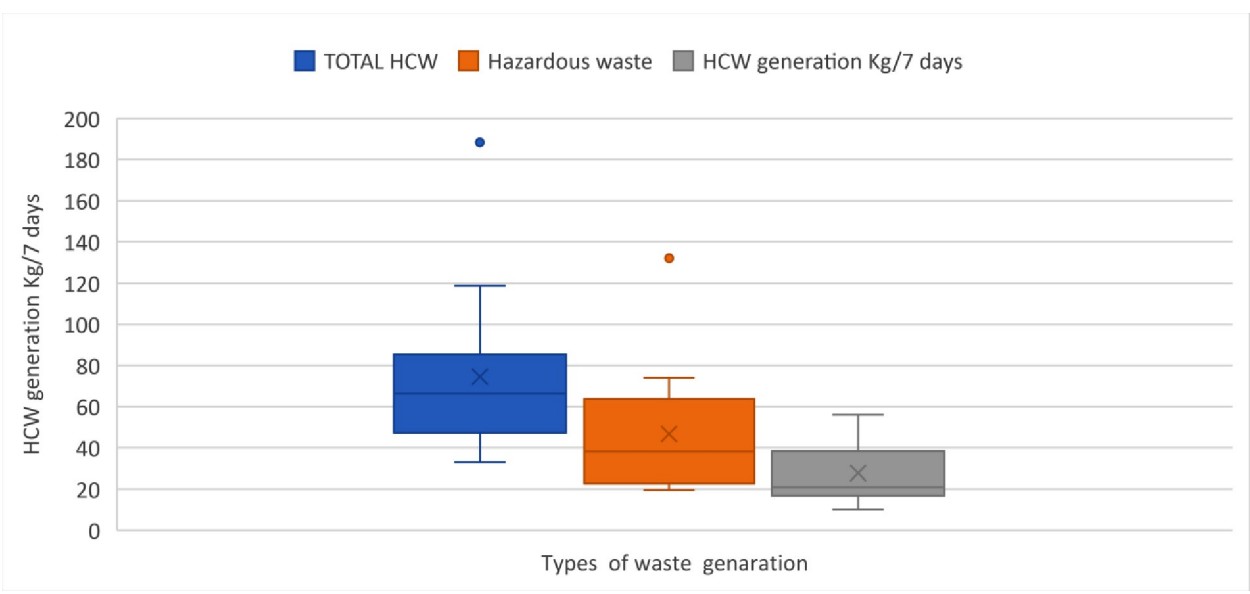

**Fig 4. Hazardous HCW generation rate (kg/day) in the study public health centres, Addis Ababa City Administration, February 2018.**

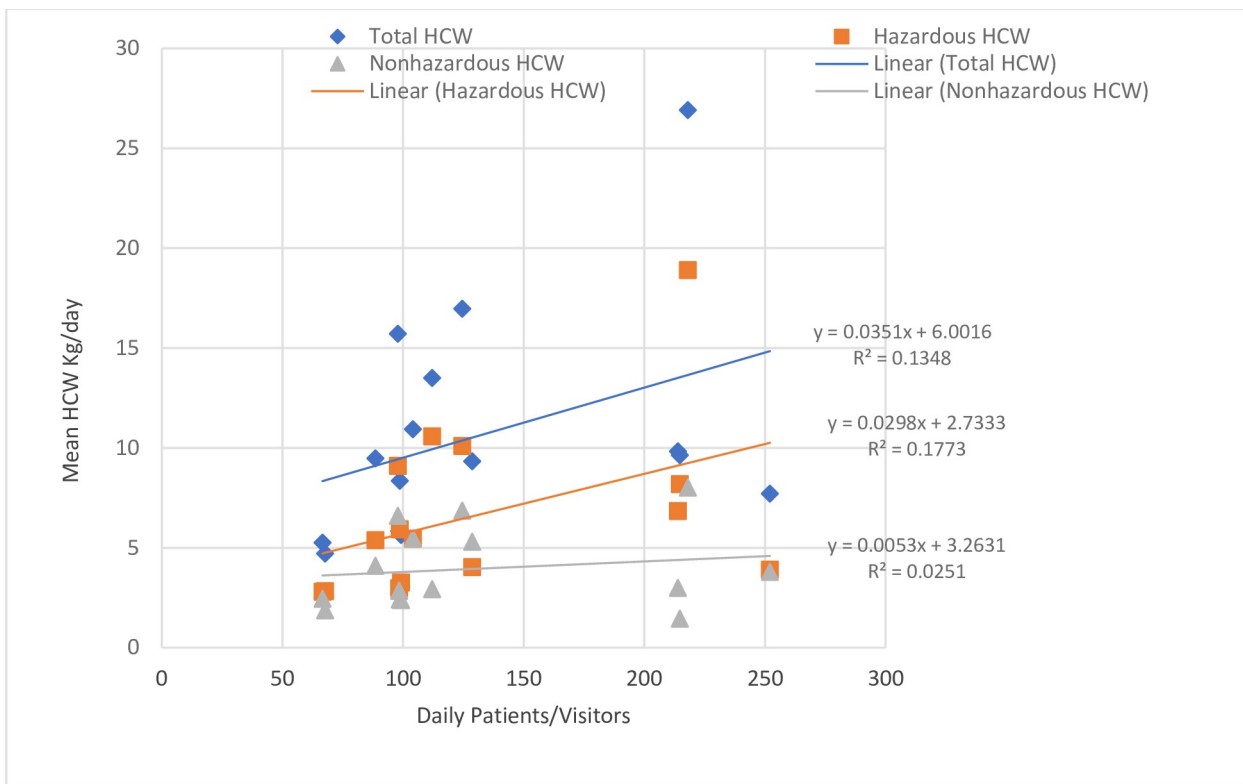

**Fig 5. Relationship between HCW generated in kg/day and the daily number of patients at the study public health centres, Addis Ababa City Administration, February 2018.**

which accounted for 7.3% of HCW [23]. This might be of the proportion of health centres per population and the seasonal variation during the study done be the factor.

The annual mean (±SD) healthcare waste generation rate estimate per health center in this study was 3870.53 ± 2109.84 kg/year. The estimation of annual mean healthcare waste generation rate can vary in the assumptions. Similar study done in Addis Ababa health centres was 3501.86 ± 1204.29 kg/year [6]. This may be due to the variation of annual patient flow and seasonal variation. However, the assumption was the preferable method to estimate annual health care waste generation rate because the mean of annual healthcare waste was determined by annual patient flow within the health center.

The mean healthcare waste generation rate in this study was 80g/patient/day per health center. It was higher than the study done in Addis Ababa, Ethiopia, health centers 57.37g/per day per patient [6]. It was also different from another study done in Jimma Zone, Jimma Medical Centre, South Western Ethiopia, the mean medical waste generation rate was 750gm/patient/day [24]. This variation may be due to geographical location, season of the year, availability of different facilities, social status of the patients (i.e., income, living standard, awareness about disease), healthcare waste management and healthcare waste legislation of the zone.

## Conclusion

The mean of healthcare waste generated in all health centres was greater in amount it needs good management practice unless it results unsafe and may have negative impact to the healthcare workers, patients and the environment. During the healthcare waste collection survey week large amount took part in health centres has no public hospitals nearby. This showed us most health centres done different procedures for their patients and generate more healthcare waste. The study findings contribute to all health centres and facilities to prepare healthcare waste management plan in order to minimize generation, promote proper segregation, collection, transportation and final disposal.

Addis Ababa City Administration Health Bureau and respective health offices and health centres need to provide vaccination for all health workers, ancillary staffs and healthcare waste handlers; provision of necessary materials, supplies and equipment and prepare protocols for measurement and quantification of healthcare waste. Effective management help the community, the healthcare workers and the environment by preventing the spreading of disease and conserving the resources.

To improve operations healthcare facilities should appoint infection control teams / committees that include specialists to occupational and environmental health and waste management experts. The findings of the study should contribute to the achievement of the United Nations [25] sustainable development goals (SDGs) for 2016–2030, which are aimed at bringing about a sustainable world and protecting the planet such as SDG3: Good health and wellbeing, SDG6: Clean water and sanitation, SDG8: Decent work and economic growth, SDG12: Responsible consumption and production and SDG13: Climate action.

## Limitation of the study

The study conducted was cross-sectional and couldn't identify causality. The study was conducted in public health centres healthcare waste generation and quantification and couldn't represent healthcare waste outside the public health centers. Private health care facilities not included in this study. The study is conducted the generation and quantification rate on solid healthcare waste other studies should also address HCW management issues is very important.

## Acknowledgments

First, I would like to express my deepest gratitude to Professor Bethabile Lovely Dolamo for her unreserved support throughout the study period. I sincerely thank University of South Africa, Menelik II Medical and Health Science College, Addis Ababa City Government Health Bureau, heads of the study health centers for their unreserved cooperation during data collection time. My deepest gratitude also goes to all data collectors and supervisors for their commitment during data collection. I would like to thank my beloved wife Alemnesh Mude, daughters Bezawit Menelik and Hermela Menelik for their patience during the study period.

## Author Contributions

**Conceptualization:** Menelik Legesse Tadesse.

**Data curation:** Menelik Legesse Tadesse.

**Formal analysis:** Menelik Legesse Tadesse.

**Methodology:** Menelik Legesse Tadesse.

**Software:** Menelik Legesse Tadesse.

**Writing – original draft:** Menelik Legesse Tadesse.

**Writing – review & editing:** Menelik Legesse Tadesse.

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
