## [Decision Letter · Decision Letter 0]

21 Mar 2023

PONE-D-22-33271Healthcare waste generation and quantification in public health centres in Addis Ababa, EthiopiaPLOS ONE

Dear Dr. Tadesse,

Thank you for submitting your manuscript to PLOS ONE. After careful consideration, we feel that it has merit but does not fully meet PLOS ONE’s publication criteria as it currently stands. Therefore, we invite you to submit a revised version of the manuscript that addresses the points raised during the review process.

We look forward to receiving your revised manuscript.

Kind regards,

Aiggan Tamene

Academic Editor

PLOS ONE

Journal Requirements:

- https://doi.org/10.1371/journal.pone.0277209

- http://dx.doi.org/10.4314/ejhs.v28i2.4

In your revision ensure you cite all your sources (including your own works), and quote or rephrase any duplicated text outside the methods section. Further consideration is dependent on these concerns being addressed.

“No funders had no role in this study. I didnt get fund for this study”

5. Please ensure that you refer to Figure 3 in your text as, if accepted, production will need this reference to link the reader to the figure.

6. Please upload a copy of Figure 6, to which you refer in your text on page 18. If the figure is no longer to be included as part of the submission please remove all reference to it within the text.

Additional Editor Comments:

Dear Author thank you for your work on this topic.

I have some comments and questions which are very important for the improvement of your paper before publishing

Abstract:

Comment #1 what is the importance of discussing the methods of health care waste disposal system although your title is on generation and quantification of HCW?

Did you assess the disposal system of HCW in this study?

Comment #2: Rewrite this sentence again there is a repetition of the words "The aim of the study was study to assess the type of healthcare waste generation and quantification from selected public health centers in Addis Ababa Ethiopia"

Comment #3: You assessed the type of healthcare waste generation. What type is it? Did you study the rate of HCW generation or the type of HCW generation?

Comment #4: Is the SPSS appropriate for data entry?

Comment #5: Based on research findings 3/4 of HCW is general waste. But you come up with 62.74% hazardous waste. What is the reason behind this finding? Can you give a justification?

Comment #6: Your objective is about waste generation rate and quantification, but your conclusion was for waste segregation and its management practice.

Comment #7: How can waste segregation improve the generation rate? Which one comes first?

Background:

Comment #1: For paragraph one where is its source? It is better to cite particularly the source.

All paragraphs did not explain the background information regarding the waste generation rate and its quantification. But it explains the practice of HCW management, its risks, and consequences for the environment and human beings. I suggest you rewrite the background and to full fill in some incomplete and nonsense paragraphs like paragraph five.

Materials and Methods:

Comment # 1: Why for you focused to study on a Health center setting only? Why not other hospitals?

Comment # 2: What by mean functional health centers? Were there any health centers nonfunctional at the study time?

Comment # 3: Your study's Inclusion and exclusion criteria are the same. Try to identify them and justify what you have done in your study.

Comment # 4: What was your role in data collection? Is it ethical for the researcher to measure the study variables?

Comment # 5: How you assured the health and safety of data collectors?

Comment # 6: How you categorized wastes as hazardous and general?

Results and Discussion:

As I have described in the part of Abstract how can segregation affect the generation rate?

Rewrite this limitation part sentence again [The study was conducted in public health centres healthcare waste generation and couldn’t represent healthcare waste outside the public health centers (private HCF)].

Conclusion:

Is it possible to conclude about the collection, segregation, and transportation and disposal system of HCW with your study objective, design, and findings?

Reviewers' comments:

Reviewer's Responses to Questions

**Comments to the Author**

1. Is the manuscript technically sound, and do the data support the conclusions?

Reviewer #1: Yes

Reviewer #2: No

2. Has the statistical analysis been performed appropriately and rigorously? 

Reviewer #1: Yes

Reviewer #2: No

3. Have the authors made all data underlying the findings in their manuscript fully available?

Reviewer #1: Yes

Reviewer #2: No

4. Is the manuscript presented in an intelligible fashion and written in standard English?

Reviewer #1: Yes

Reviewer #2: No

5. Review Comments to the Author

Reviewer #1: Comment 1:- under the abstract section of background “Modern methods to dispose of healthcare waste have been introduced to most healthcare institution and yet there is evidence of healthcare waste mismanagement with reference to public health centres in Ethiopia” be written like “Healthcare waste management have been introduced to all healthcare institution within the country, yet there is evidence of poor healthcare waste management practice in public health centres in Ethiopia” or you can write it in other form.

Comment 2:- under the abstract section of background “The aim of the study was study to assess …….” Remove “study”

Comment 3:- under the abstract section of result “the mean (±SD) HCW generation rate in each section was 10.63±5.795 kg/day” is repeated.

Comment 4:- I don’t think it is appropriate to start your background with a paragraph like this “Healthcare waste management is very important due to its hazardous nature …….healthcare waste mismanagement with reference to public health centres in Ethiopia.” So you better remove it from the background”

Comment 5:- The sampling technique and sample size determination methods are not clearly stated.

Comment 6:- under the data collection tool and procedure section, you better clearly state how you prevent mixing of waste of different categories at the point of generation. (Your result “62.7% hazardous waste” showed how the segregation was poor)

Comment 7:- under result section of patient flow in the study health centers you better rewrite the narrative paragraph.

Comment 8:- What type of wastes did you consider as general waste? If possible, please classify general waste as plastic, garden wastes, papers etc.

Comment 9:- in the result section, You have used the unit gram and Kilogram at different parts for single variable like HCW generation rate per patient per day, so I suggest you better use either of one uniformly.

Comment 10:- the discussion and conclusion part has significant grammatical and structural problems, therefore you better revise these parts wholly.

Reviewer #2: While the document contains important strong points, there are significant issues with the methodological approach used by the author/s. I call on the author to review and resubmit after considering the comments forwarded

6. PLOS authors have the option to publish the peer review history of their article (what does this mean?). If published, this will include your full peer review and any attached files.

Reviewer #1: No

Reviewer #2: No

---

## [Author Response · Author response to Decision Letter 0]

10 Jul 2023

Dear reviewers it is my great pleasure to submit the revised manuscript hope you will see it soon and ready for publication.

---

## [Decision Letter · Decision Letter 1]

7 Aug 2023

PONE-D-22-33271R1Healthcare waste generation and quantification in public health centres in Addis Ababa, EthiopiaPLOS ONE

Dear Dr. Tadesse,

Thank you for submitting your manuscript to PLOS ONE. After careful consideration, we feel that it has merit but does not fully meet PLOS ONE’s publication criteria as it currently stands. Therefore, we invite you to submit a revised version of the manuscript that addresses the points raised during the review process. Please submit your revised manuscript by Sep 21 2023 11:59PM. If you will need more time than this to complete your revisions, please reply to this message or contact the journal office at plosone@plos.org. Please include the following items when submitting your revised manuscript:A rebuttal letter that responds to each point raised by the academic editor and reviewer(s). You should upload this letter as a separate file labeled 'Response to Reviewers'.A marked-up copy of your manuscript that highlights changes made to the original version. You should upload this as a separate file labeled 'Revised Manuscript with Track Changes'.An unmarked version of your revised paper without tracked changes. You should upload this as a separate file labeled 'Manuscript'.

We look forward to receiving your revised manuscript.

Kind regards,

Aiggan Tamene

Academic Editor

PLOS ONE

Reviewers' comments:

Reviewer's Responses to Questions

**Comments to the Author**

1. If the authors have adequately addressed your comments raised in a previous round of review and you feel that this manuscript is now acceptable for publication, you may indicate that here to bypass the “Comments to the Author” section, enter your conflict of interest statement in the “Confidential to Editor” section, and submit your "Accept" recommendation.

Reviewer #3: All comments have been addressed

Reviewer #4: (No Response)

2. Is the manuscript technically sound, and do the data support the conclusions?

Reviewer #3: Partly

Reviewer #4: Partly

3. Has the statistical analysis been performed appropriately and rigorously? 

Reviewer #3: No

Reviewer #4: Yes

4. Have the authors made all data underlying the findings in their manuscript fully available?

Reviewer #3: No

Reviewer #4: Yes

5. Is the manuscript presented in an intelligible fashion and written in standard English?

Reviewer #3: No

Reviewer #4: No

6. Review Comments to the Author

Reviewer #3: The manuscript is interesting and good in scientific content and has novelty. It is well fit with the scope and aim of the journal. However, I recommend that a major revision is warranted. A more detailed review can be found in the specific comments below. I ask that the authors specifically address each of my comments in their response.

Abstract:

#Abstract is written good but could you revise the background information. It should only key research gap on the topic.

Reviewing the submitted manuscript without adding the line numbers is very difficult. Authors must add the line number before submitting the revised version of the manuscript, if any. thus, I commented directly in the attached PDF through annotation tools.

Introduction: There are numerous paragraphs through the introduction. Authors are advised the provide complete introduction within 4-5 paragraphs break.

#Introduction needs more clarity with state of arts, objectives of the study for the scientific novelties of manuscript. #Additionally, the introduction should cover the recent literature related to this subject. Introduction is completely lacking with the citation from the years 2022; 2023.

#What are the differences between this study and others in the literature? The originality/novelty of the paper should be clearly stated in the introduction section.

#Please follow the literature review by a clear and concise state-of-the-art analysis. This should clearly show the knowledge gaps identified and link them to your paper goals. Please reason both the novelty and the relevance of your paper goals.

#Provide one nice and technically sound paragraph at the end of introduction section about what is covered in the manuscript. Before it also adds one section on Knowledge gaps in the introduction.

#In the statistical analysis section, the details of used software i.e. name, versions, and make must be mentioned.

I did not find any explanation about the importance of using the statistical analysis approach in the study

Discussion

#The discussion section still needs improvement and should be linked to the findings of the previous reports on this topic. discussion is elaborative but it needs more adequate discussion with supporting latest references. Discussion should be according to the results. Authors should state each citation to its specific discovery.

#I suggest reworking the discussion in order to more sufficiently frame the theoretical, empirical, policy and methodological concerns in the paper.

#The major defect of this study is the debate or argument on the significance of the work is not clearly stated in the introduction section. Hence, the contribution is weak in this manuscript. I would suggest the authors enhance the discussion to justify the novelty.

#Replace the older references with recent literature (2020-2023) in the discussion sections.

Refences:

#For citations and reference within the text, author must follow guide for authors. The references must be also in the format of the journal.

Others

#Unfortunately, the manuscript contains numerous typos, stylistic issues, and some grammatical errors. Manuscript should be checked once for any grammatical as well as typological errors like somewhere spaces and comma (,), if not given, this should be corrected.

#Table legends, figure captions, and foot notes need improvement. All legends, captions, and foot notes should have enough description for a reader to understand the figure without having to refer back to the main text of the manuscript. Avoid the use of abbreviation in figure legends.

All the sections of the manuscript should be revised properly. The manuscript requires major revision.

Reviewer #4: 1. The research gap and objective of the present study is missing in the “Background” section.

2. “Background” of any research should contain Research Background and Context, Research Question or Objective, Brief Literature Review, Significance and Purpose, Methodology, Outline of the Paper. Revised this section based on the above points.

3. Discussion is poorly written even after revision. Though reviewers have given their on discussion to improve but authors fail to incorporate it.

4. I would like to give an example regarding “Comment #5: Based on research findings 3/4 of HCW is general waste. But you come up with 62.74% hazardous waste. What is the reason behind this finding? Can you give a justification?”. In the response of this comment’s authors replied that “there is no segregation at the point of

5. Generation” but it is very surprising that authors did not provide any justification in the revised manuscript.

6. A thorough and extensive revision is required in the discussion is required which will not only based on the comparison of the result of the present study with existing literature but also authors have to justify the reason of getting this type of results which they have obtained.

7. Detail description on “Study design” is required with proper references.

8. These section “Study units”,“Source population”, “Study units” are not understandable. Author should rewrite this section with detail information.

9. It seems that authors has follows the own devolved methodology for the current study as Methodology section is badly suffering form proper citation.

10. Some equations are not readable also.

11. Conclusion is the crux of any paper. The conclusion of a research paper should typically include Summary of Findings, Interpretation of Results, Contribution to Knowledge, Limitations, Recommendations, Closing Thoughts. Remember to keep the conclusion focused, concise, and well-structured, leaving a lasting impression on the reader while summarizing the significance of your work. Rewrite the conclusion based on the above scenario.

12. Author should improve the English language.

7. PLOS authors have the option to publish the peer review history of their article (what does this mean?). If published, this will include your full peer review and any attached files.

Reviewer #3: No

Reviewer #4: No

---

## [Author Response · Author response to Decision Letter 1]

20 Sep 2023

Dear Reviewers, nice to see you. As your comment I upload the revised documents as much as possible. I would like to thank you all for your constructive comments and I will see your response soon.

---

## [Decision Letter · Decision Letter 2]

17 Nov 2023

Healthcare waste generation and quantification in public health centres in Addis Ababa, Ethiopia

PONE-D-22-33271R2

Dear Dr. Tadesse,

We’re pleased to inform you that your manuscript has been judged scientifically suitable for publication and will be formally accepted for publication once it meets all outstanding technical requirements.

Kind regards,

Sylvester Chidi Chima, M.D., L.L.M, LLD.

Academic Editor

PLOS ONE

**Comments to the Author**

1. If the authors have adequately addressed your comments raised in a previous round of review and you feel that this manuscript is now acceptable for publication, you may indicate that here to bypass the “Comments to the Author” section, enter your conflict of interest statement in the “Confidential to Editor” section, and submit your "Accept" recommendation.

Reviewer #1: All comments have been addressed

Reviewer #3: All comments have been addressed

2. Is the manuscript technically sound, and do the data support the conclusions?

Reviewer #1: Yes

Reviewer #3: Yes

3. Has the statistical analysis been performed appropriately and rigorously? 

Reviewer #1: Yes

Reviewer #3: Yes

4. Have the authors made all data underlying the findings in their manuscript fully available?

Reviewer #1: Yes

Reviewer #3: (No Response)

5. Is the manuscript presented in an intelligible fashion and written in standard English?

Reviewer #1: Yes

Reviewer #3: Yes

6. Review Comments to the Author

Reviewer #1: I would like to appreciate the author for your dedication. Almost all my comments been addressed and amendments had been made as per my comment.

Reviewer #3: (No Response)

7. PLOS authors have the option to publish the peer review history of their article (what does this mean?). If published, this will include your full peer review and any attached files.

Reviewer #1: **Yes: **Abel Artwork

Reviewer #3: No

---

## [Editor Report · Acceptance letter]

26 Jan 2024

PONE-D-22-33271R2 

PLOS ONE

Dear Dr. Tadesse, 

I'm pleased to inform you that your manuscript has been deemed suitable for publication in PLOS ONE. Congratulations! Your manuscript is now being handed over to our production team.

Kind regards, 

on behalf of

Professor Sylvester Chidi Chima 

Academic Editor

PLOS ONE